# Impact of Multiple Sclerosis on Load Distribution, Plantar Pressures, and Ankle Dorsiflexion Range of Motion in Women

**DOI:** 10.3390/healthcare13111231

**Published:** 2025-05-23

**Authors:** Sara Zúnica-García, Esther Chicharro-Luna, Alba Gracia-Sánchez, Isabel Jiménez-Trujillo, Jonatan García-Campos, Ángel P. Sempere

**Affiliations:** 1Department of Behavioral Sciences and Health, Nursing Area, Faculty of Medicine, Miguel Hernández University, 03550 Alicante, Spain; szunica@umh.es (S.Z.-G.); ec.luna@umh.es (E.C.-L.); agracia@umh.es (A.G.-S.); 2Institute of Health and Biomedical Research of Alicante (ISABIAL), 03010 Alicante, Spain; angel.perezs@umh.es; 3Department of Medical Specialties and Public Health, Health Sciences Faculty, Rey Juan Carlos University, 28008 Madrid, Spain; isabel.jimenez@urjc.es; 4Department of Neurology, Dr. Balmis General University Hospital, 03010 Alicante, Spain; 5Department of Clinical Medicine, Miguel Hernández University, 03550 Alicante, Spain

**Keywords:** multiple sclerosis, plantar load, plantar pressure, pressure platform, static

## Abstract

Alterations in static plantar pressure distribution serve as important indicators of gait and balance impairments in individuals with Multiple Sclerosis (MS). In addition, the identification of altered patterns of plantar load distribution, along with restricted ankle dorsiflexion, may serve as early markers of postural instability and gait dysfunction in women with MS. **Objectives**: To assess differences in static plantar pressure, load distribution, and ankle dorsiflexion range of motion between women diagnosed with MS and women without the condition. **Methods**: A cross-sectional observational study was conducted between April and December 2024. Women with MS were recruited from patient associations in the provinces of Alicante and Murcia, as well as from the neurology outpatient clinic at the Doctor Balmis University Hospital (Alicante, Spain). Static postural assessment was performed using the Neo-Plate^®^ pressure platform, which measured maximum and mean plantar pressure (kPa), load distribution (%), contact surface area (cm^2^), and anterior–posterior weight distribution between the forefoot and rearfoot. The ankle dorsiflexion range of motion was assessed with a universal two-arm goniometer. All parameters were compared with those of a group of women without a diagnosis of MS. **Results**: Compared to women without MS, participants with MS showed a significantly greater load on the right forefoot (25.75% vs. 23.41%, *p* = 0.021), and reduced load on the right (23.09% vs. 26.01%, *p* = 0.004) and left rearfoot (26.60% vs. 30.85%, *p* = 0.033). Total forefoot loading was significantly higher (52.33% vs. 46.40%, *p* < 0.001), and rearfoot loading was lower (47.64% vs. 52.42%, *p* = 0.006) in the MS group. Ankle dorsiflexion range of motion was also significantly reduced in women with MS, both with the knee flexed (5.95° ± 4.50 and 6.76° ± 4.69 vs. 15.45° ± 5.04 and 14.90° ± 5.43) and extended (2.69° ± 3.69 and 3.12° ± 3.83 vs. 8.17° ± 3.41 and 8.60° ± 3.31), with all differences reaching statistical significance (*p* < 0.001). **Conclusions**: Women with MS present significant alterations in static plantar load distribution, with increased forefoot and decreased rearfoot loading, as well as markedly reduced ankle dorsiflexion, in comparison to women without the disease. These findings suggest the presence of postural imbalances associated with MS, potentially affecting functional stability and mobility.

## 1. Introduction

Multiple sclerosis (MS) is a chronic autoimmune disease of the central nervous system, characterized by inflammation, demyelination, and progressive neurodegeneration [1,2]. It predominantly affected women, with a reported female-to-male ratio of approximately 2.1:1 [3]. Significant sex-related differences in disease progression and inflammatory activity had been described [4]. This higher prevalence among women was thought to be associated with genetic, hormonal, or environmental factors, potentially influencing the clinical manifestations and course of the disease [4,5]. In Spain, the prevalence of MS has been estimated at 123.5 cases per 100,000 person-years [3].

The clinical presentation of MS is highly variable and includes motor, sensory, and cognitive impairments that affect the functionality and quality of life [6]. Furthermore, the presence of comorbidities imposes additional economic, healthcare, and social burdens [7], particularly relevant in women due to the greater impact of the disease in this population. One of the primary functional consequences of MS is postural control impairment, often linked to deficits in motor coordination, spasticity, and proprioceptive dysfunction [8]. These factors can compromise balance, alter static plantar weight distribution, and increase the risk of falls [9], thereby limiting mobility and functional independence.

In the lower limbs, and more specifically in the musculature involved in ankle mobility, individuals with MS exhibit altered activation of the tibialis anterior and triceps surae muscles [10], as well as shortening of the gastrocnemius muscle [11]. These alterations lead to a reduction in ankle joint range of motion, which some authors have interpreted as a potential compensatory strategy to enhance weight-bearing support and balance during the stance phase of gait [10]. However, this relationship is not clearly established, as several studies have examined the association between ankle dorsiflexion and balance, finding that limited dorsiflexion is associated with poorer dynamic balance [12,13]. Furthermore, this joint limitation has been linked to increased forefoot plantar pressures during gait [14,15].

Alterations in static plantar pressure distribution serve as important indicators of gait and balance impairments in individuals with MS. Plantar pressure analysis is a useful method for assessing gait and postural control, offering valuable insights into the progression of the disease and functional capacity [16,17]. Previous studies have demonstrated that sensory loss, reduced vibratory perception, and muscular spasticity, frequently observed in MS, are associated with abnormal plantar pressure patterns [16,18].

Moreover, sex-specific biomechanical and physiological factors in women may significantly influence plantar pressure, load distribution, and ankle mobility. Characteristics such as a more anterior center of gravity [19], increased knee valgus [20], greater ligamentous laxity [21], and morphological differences in the foot [22], such as a higher and stiffer plantar arch, directly affect the biomechanics of the foot-ankle complex. In the specific context of MS, sex-related differences in static postural control have also been reported, with women showing poorer performance [23]. Therefore, focusing the analysis exclusively on women allows for better control of these variables and a more in-depth understanding of the functional impairments most representative of this population, which also exhibits a higher prevalence of the disease.

The identification of altered plantar load distribution patterns, together with restricted ankle dorsiflexion, may serve as early markers of postural instability and gait dysfunction in women with MS, which in turn enables a more accurate assessment of functional status and supports the implementation of individualized rehabilitation interventions aimed at improving clinical care. The objective of this study was to analyze differences in static plantar pressure, load distribution, and ankle dorsiflexion between women diagnosed with MS and women without the disease.

## 2. Materials and Methods

### 2.1. Study Design

A cross-sectional observational study was conducted involving individuals with and without a diagnosis of MS. MS patients were recruited through four MS patient associations from the provinces of Alicante and Murcia, as well as from the neurology outpatient clinic at the Doctor Balmis University Hospital in Alicante. Control participants were recruited from a consumer association in Alicante. The recruitment and data collection period took place between April and December 2024. The study was approved by the Ethics Committee of the Alicante Institute for Health and Biomedical Research (ISABIAL), under reference number PI2024-007. The study was designed and conducted in accordance with the STROBE guidelines for observational studies [24].

### 2.2. Sample Size

The sample size was calculated using software developed by the Clinical Epidemiology and Biostatistics Unit of the University Hospital Complex of A Coruña, Universidade da Coruña (www.fisterra.com, accessed on 16 February 2025). A two-tailed test was applied with a 95% confidence level and 80% statistical power. An effect size of 5 g/cm^2^ and a standard deviation of 7.69 g/cm^2^ were used as parameters [25]. The estimated minimum sample size was 37 participants per group. Considering a potential 10% dropout rate, the final sample size was adjusted to 41 participants per group.

### 2.3. Study Population

MS patients were consecutively selected if they met the following inclusion criteria: confirmed diagnosis of MS by a neurologist [26], minimum age of 18 years, a maximum score of 5.5 on the Expanded Disability Status Scale (EDSS) [27] and signed informed consent. Exclusion criteria included the presence of other neurodegenerative diseases, history of lower limb surgery, musculoskeletal injuries within the previous six months, cognitive impairment, or psychiatric disorders that could interfere with questionnaire comprehension. Control participants were selected using the same inclusion and exclusion criteria, except for the absence of MS diagnosis.

### 2.4. Data Collection

Data were collected on sex, age (years), educational level (no formal education, primary, secondary, or university), marital status (single, married, divorced/separated, or widowed), and employment status (employed, unemployed, on medical leave, or retired/pensioner). Additionally, weight (kg), height (m), and body mass index (BMI) were recorded. BMI was categorized according to the World Health Organization (WHO) criteria as underweight (BMI < 18.50), normal weight (BMI 18.50–24.99), overweight (BMI 25–29.9), and obesity (BMI ≥ 30). Smoking habits were documented based on whether the participant was a current smoker, former smoker, or non-smoker. Physical activity level and presence of comorbidities were also recorded. For MS patients, data were also collected on years since diagnosis, age at diagnosis, clinical presentation type (relapsing-remitting, primary progressive, or secondary progressive), and EDSS score.

Static plantar pressure data collection was carried out using the Neo-Plate pressure platform (Sensor Medica, Guidonia Montecelio, Italy), which was connected to a computer managing data acquisition through its dedicated software. The platform features an active surface of 40 × 40 cm, a thickness of 8 mm, and is equipped with 1600 resistive-type sensors. Its image acquisition frequency is configurable, ranging from 100 to 500 Hz. Measurements were taken with participants standing barefoot in a static bipedal position, maintaining their natural gait angle and base of support (Figure 1a). The recorded parameters included maximum and mean plantar pressures (kPa), plantar load distribution (%) between the forefoot and rearfoot, and the contact surface area (cm^2^) of each foot, as well as the total plantar load distribution (%) between forefoot and rearfoot (Figure 1b).

Ankle joint range of motion was assessed using a standard two-arm universal goniometer. Dorsiflexion was measured with the participant in a supine position on an examination table. The goniometer’s axis was aligned with the lateral malleolus, the fixed arm was aligned with the fibula, and the movable arm was aligned with the fifth metatarsal [28].

### 2.5. Statistical Analysis

Statistical analysis was performed using SPSS^®^ software, v. 29.0 (SPSS Inc., Chicago, IL, USA). Qualitative variables were described using absolute frequencies and percentages, while quantitative variables were expressed as means and standard deviations. Normality was assessed using the Shapiro–Wilk test (n < 50).

Inferential analyses were conducted using the chi-squa.re test for categorical qualitative variables. For quantitative variables, the choice of test depended on the distribution and homogeneity of variances: the Mann–Whitney U test or Kruskal–Wallis test was used for non-parametric data, while Student’s *t*-test or ANOVA was applied when parametric assumptions were met.

Relationships between quantitative variables were examined using Pearson’s correlation coefficient when both variables followed a normal distribution, and Spearman’s rank correlation coefficient when this assumption was not met.

To evaluate the association between the dependent variable (MS) and independent variables while controlling for potential confounders, multiple logistic regression models were constructed. A backward elimination method based on maximum likelihood estimation was employed, initially including variables that showed statistical significance in the bivariate analysis (*p* < 0.05). A *p*-value < 0.05 was considered statistically significant.

## 3. Results

The study included 42 women with MS and 42 women without MS, with no statistically significant differences in mean age between groups (55.31 ± 8.27 years vs. 55.21 ± 8.07 years; *p* = 0.911). Among women with MS, the mean duration of the disease was 16.56 ± 7.68 years, and the average age at diagnosis was 38.67 ± 8.82 years. The disability level, assessed using the EDSS, showed a mean score of 2.55 ± 1.50, indicating mild to moderate functional impairment. Regarding clinical phenotype, the majority of women with MS (76.2%) were classified as having relapsing–remitting MS, while 2.4% had primary progressive MS and 19% had secondary progressive MS. A significantly higher proportion of former smokers was observed among women with MS compared to women without MS (52.4% vs. 31%; *p* = 0.030), potentially reflecting differences in historical tobacco use. Emotional disorders were also more prevalent in women with MS, with significantly higher rates of depression (33.3% vs. 7.1%; *p* = 0.003) and anxiety (38.1% vs. 11.9%; *p* = 0.006) compared to women without MS (Table 1).

In terms of static plantar load distribution, women with MS exhibited a significantly greater load on the right forefoot (25.75% vs. 23.41%; *p* = 0.021). In contrast, lower values were recorded in this group on the right rearfoot (23.09% vs. 26.01%; *p* = 0.004) and left rearfoot (26.60% vs. 30.85%; *p* = 0.033) compared to women without MS. Total forefoot load was also significantly higher in women with MS (52.33% vs. 46.40%; *p* < 0.001), while total rearfoot load was lower (47.64% vs. 52.42%; *p* = 0.006). These findings indicate a shift in plantar pressure distribution in women with MS, which may contribute to reduced postural stability (Table 2).

Women with MS showed a significantly reduced ankle dorsiflexion range of motion compared to those without the condition. With the knee flexed, they recorded mean values of 5.95° ± 4.50 in the right limb and 6.76° ± 4.69 in the left, whereas the control group achieved 15.45° ± 5.04 and 14.90° ± 5.43, respectively. This difference persisted with the knee extended, with values of 2.69° ± 3.69 and 3.12° ± 3.83 in the MS group, compared to 8.17° ± 3.41 and 8.60° ± 3.31 in the control group. All differences were statistically significant (*p* < 0.001) (Table 3).

Additionally, associations between participant characteristics and the main outcome measures were explored. BMI showed significant correlations with several outcomes: ankle dorsiflexion with the knee flexed on the left foot (rs = −0.254; *p* = 0.020), mean plantar pressure in both feet (right: rs = 0.284; *p* = 0.009; left: rs = 0.331; *p* = 0.002), plantar contact surface area (right: rs = 0.686; *p* < 0.001; left: rs = 0.601; *p* < 0.001), forefoot load on the left foot (rs = 0.229; *p* = 0.036), and total forefoot and rearfoot load (rs = −0.230; *p* = 0.035 and rs = 0.225; *p* = 0.019, respectively). MS clinical phenotype was also significantly associated with mean plantar pressure in the left foot (*p* = 0.024), with higher values observed in participants with relapsing–remitting MS (144.02 ± 174.95 kPa), followed by those with primary progressive MS (114.50 ± 0 kPa) and secondary progressive MS (103.86 ± 7.76 kPa). No other significant associations were found between age, disease duration, or EDSS and the remaining outcome measures.

Finally, multivariate analysis showed a statistical association between MS diagnosis and anxiety, forefoot load, and ankle dorsiflexion in different conditions. The model showed good explanatory power, with a Nagelkerke R^2^ value of 0.75 (Table 4).

## 4. Discussion

The objective of this study was to analyze differences in static plantar pressure, load distribution, and ankle dorsiflexion between women diagnosed with MS and women without the disease. The results indicated that individuals with MS exhibit greater forefoot loading and reduced rearfoot loading compared to controls. From a biomechanical perspective, this pattern may reflect an anterior shift in the center of gravity, which could compromise postural stability and increase the risk of falls, particularly during dynamic activities such as walking or positional transitions.

Our findings are consistent with those reported by Erdeo et al. [25], who identified significant differences in plantar load distribution among individuals with MS, particularly in the lateral forefoot, lateral rearfoot, and medial rearfoot regions. Additionally, they noted that patients with coordination impairments exhibited greater lateral rearfoot loading compared to those with balance-related issues. This suggests that motor deficits in MS may differentially influence plantar load distribution, potentially explaining the altered postural control observed in our study. Furthermore, our findings regarding plantar load distribution in women with MS align with previous studies by Balgetir et al. [29] and Kaya et al. [17], which demonstrated that plantar pressure analysis using deep learning models can detect early-stage ataxic patterns in MS. These findings support the clinical relevance of assessing plantar load in MS patients, as it may enable more accurate detection of postural dysfunctions and inform targeted interventions to improve stability and mobility.

Another relevant factor in postural control alteration in MS is spasticity, which has been identified as a key contributor to gait and balance impairments. Norbye et al. [30] demonstrated that even low levels of spasticity in the ankle plantar flexors, knee extensors, and hip adductors negatively impact walking ability and balance. Our results show a significant reduction in the dorsiflexion range of motion of the ankle compared to the control group. This aligns with previous research demonstrating that altered activation of the tibialis anterior and gastrocnemius muscles leads to a reduction in ankle joint range of motion [10]. Additionally, the assessment of dorsiflexion with the knee flexed showed an increase in the range of motion in both groups, which is again consistent with the reduction in gastrocnemius muscle tension when the knee is flexed, allowing for greater ankle dorsiflexion [10]. In this context, the altered plantar load distribution observed in our study may be influenced by spasticity in these muscle groups, and the reduction in ankle dorsiflexion, contributing to increased forefoot loading and, consequently, reduced postural stability. Although BMI was significantly associated with several biomechanical variables, no differences in BMI were observed between women with and without MS. This suggests that the observed associations are more likely attributable to individual variability rather than to the diagnosis itself. Therefore, BMI should be considered an independent factor when interpreting these results.

Finally, our results showed significantly higher rates of anxiety and depression among women with MS, which may have a direct impact on postural control. Feldman et al. [31] found that anxiety in the general population is associated with gait disturbances such as reduced speed, shorter step length, and lower cadence, as well as deficits in balance and mobility. Similarly, in patients with Parkinson’s disease, anxiety increases center of gravity variability and affects body inclination, further compromising stability [32]. These findings suggest that anxiety may be a contributing factor to postural instability in women with MS.

This study presents several limitations that should be considered. The sample size was relatively small, consisting of 84 women with and without MS, which may restrict the generalizability of the findings to broader populations. Moreover, the sample was composed exclusively of women and, thus, the results cannot be extrapolated to the male population, who may exhibit different biomechanical patterns in plantar load distribution. Plantar pressure measurements were conducted only under static conditions, without assessing dynamic activities such as walking or other functional movements. This limits the applicability of the results to real-life contexts, especially considering the progressive impact of MS on mobility. Additionally, participants had EDSS scores up to 5.5, indicating mild to moderate disability without the need for walking aids; therefore, the findings may not be representative of individuals with more severe disability levels. Finally, important clinical variables such as medication use, fatigue, and pain were not controlled, all of which could have influenced plantar load distribution and postural stability. Future research with larger samples, dynamic assessments, and control of these clinical variables will provide a more comprehensive understanding of the effects of MS on postural stability.

In this regard, incorporating dynamic assessments of plantar pressure in future research would be highly beneficial. Previous studies have shown that individuals with MS present significant alterations in dynamic plantar pressure parameters during gait. For instance, Keklicek et al. [16] reported disrupted rearfoot loading and delayed load transfer to the forefoot in people with MS, indicating impaired coordination in the gait cycle. Similarly, Jones and van Emmerik [18] found that individuals with reduced plantar vibration sensitivity exhibited increased plantar pressures during walking, particularly in the less sensitive foot. These findings suggest that altered sensory input may contribute to compensatory pressure-loading strategies during gait. Integrating dynamic plantar pressure analysis with static assessments could provide a more comprehensive under-standing of postural and gait adaptations in MS and inform more effective rehabilitation strategies.

Taken together, these results underscore the importance of incorporating plantar load distribution assessment as a fundamental component of balance and gait evaluation in MS patients. Early identification of abnormal plantar loading patterns may facilitate the development of targeted therapeutic strategies aimed at improving postural stability, reducing fall risk, and optimizing mobility in this population.

### Implications for Clinical Practice

The findings of this study emphasize the importance of incorporating the assessment of ankle joint mobility and the analysis of plantar pressure distribution in women with MS into daily clinical practice. These alterations, which may initially be underestimated, can have a direct impact on postural stability, acting as risk factors for falls. Therefore, the results obtained from incorporating ankle mobility assessments and plantar pressure analysis can guide the design of early, individualized rehabilitation interventions, within a multidisciplinary care approach, aimed at improving ankle function, preventing potential complications, and enhancing the quality of life for these individuals.

Among the evidence-based strategies are eccentric exercises, which have been shown to be effective in improving ankle joint mobility [33]. Likewise, the use of dynamic foot orthoses [34], textured insoles [35], and footwear with additional cushioning [36] has demonstrated benefits for balance and gait. In addition, robot-assisted sensorimotor ankle training appears to be a promising tool to enhance strength, range of motion, balance, and locomotion in this population [37].

Although this study does not focus on investigating the underlying causes of limitation in ankle dorsiflexion, it has been identified that various conditions can contribute to this alteration, such as soft tissue contractures, joint blocks, and neurological alterations, among others [38]. Once the limitation has been detected in this population, it is necessary for future studies to address the possible underlying causes, with the aim of developing guidelines and protocols that guide personalized therapeutic interventions, considering sex differences.

## 5. Conclusions

Patients with MS exhibit increased forefoot loading and decreased rearfoot loading in static standing posture compared to individuals without the disease. In addition to a reduced ankle dorsiflexion range compared to healthy controls, both with the knee in extension and in flexion. These findings confirm alterations in static plantar load distribution in women with MS, potentially related to structural and functional changes associated with the disease. The results highlight the importance of incorporating plantar load distribution and ankle joint mobility assessment into the clinical evaluation of postural stability in this population. Early detection of abnormal loading patterns could facilitate the development of targeted interventions aimed at improving balance, reducing fall risk, and enhancing mobility.

## Figures and Tables

**Figure 1 healthcare-13-01231-f001:**
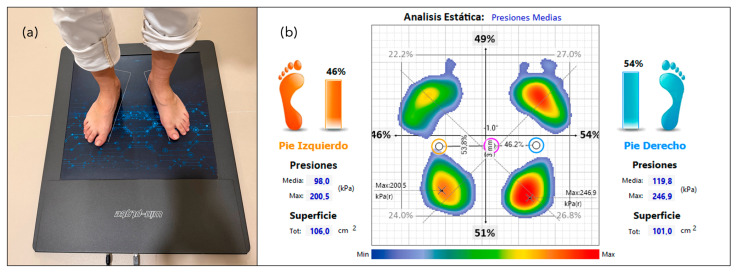
Static plantar pressure data collection. (**a**) Pressure platform; (**b**) static plantar pressure.

**Table 1 healthcare-13-01231-t001:** Sociodemographic characteristics in patients with and without MS.

Sociodemographic Characteristics	Patients with MS n = 42	Patients Without MS n = 42	*p* Value
	Mean ± SD n (%)	Mean ± SD n (%)	
Age (years)	55.31 ± 8.27	55.21 ± 8.07	0.911 ^a^
BMI (kg/m^2^)			0.224
Underweight (BMI < 18.5)	2 (4.8)	0 (0)	
Normal weight (BMI 18.5–24.9)	17 (40.5)	24 (57.1)	
Overweight (BMI 25–29.9)	14 (33.3)	13 (31)	
Obesity (BMI ≥ 30)	9 (21.4)	5 (11.9)	
BMI (kg/m^2^)	24.65 ± 3.89	26.09 ± 5.80	0.357 ^a^
Educational level			0.114
No formal education	1 (2.4)	1 (2.4)	
Primary education	10 (23.8)	6 (14.3)	
Secondary education	18 (42.9)	11 (26.2)	
University education	13 (31)	24 (57.1)	
Marital status			0.757
Single	9 (21.4)	7 (16.7)	
Married	24 (57.1)	27 (64.3)	
Separated or divorced	5 (11.9)	6 (14.3)	
Widowed	4 (9.5)	2 (4.8)	
Employment status			0.093
Employed	19 (45.2)	30 (71.4)	
Unemployed	6 (14.3)	2 (4.8)	
On medical leave	4 (9.5)	3 (7.1)	
Retired or receiving a pension	13 (31)	7 (16.7)	
Smoking habits			0.030 *
Smoker	8 (19)	5 (11.9)	
Former smoker	22 (52.4)	13 (31)	
Non-smoker	12 (28.6)	24 (57.1)	
Physical activity			0.794
Sedentary	10 (23.8)	9 (21.4)	
Physically active	32 (76.2)	33 (78.6)	
Comorbidities			
Diabetes mellitus	4 (9.5)	1 (2.4)	0.316
Hypothyroidism	7 (16.7)	6 (14.3)	0.763
Arterial hypertension	6 (14.3)	8 (19)	0.558
Hypercholesterolemia	13 (31)	11 (26.2)	0.629
Irritable bowel syndrome	1 (2.4)	3 (7.1)	0.616
Depression	14 (33.3)	3 (7.1)	0.003 *
Anxiety	16 (38.1)	5 (11.9)	0.006 *

Abbreviations: BMI, body mass index; MS, multiple sclerosis; SD, standard deviation. Chi-square test. ᵃ Mann–Whitney U test. * Statistical significance at *p* < 0.05.

**Table 2 healthcare-13-01231-t002:** Pressure platform: static analysis in patients with and without MS.

Pressure Platform	Patients with MS n = 42	Patients Without MS n = 42	*p* Value
	Mean ± SD	Mean ± SD	
Mean pressure of the right foot (kPa)	109.28 ± 9.18	111.12 ± 8.63	0.345 ^b^
Mean pressure of the left foot (kPa)	111.72 ± 9.34	111.72 ± 7.24	0.999 ^b^
Maximum pressure of the right foot (kPa)	232.87 ± 14.64	234.77 ± 13.46	0.390 ^a^
Maximum pressure of the left foot (kPa)	240.18 ± 13.19	242.21 ± 9.62	0.750 ^a^
Surface area of the right foot (cm^2^)	115.43 ± 24.38	111.07 ± 22.42	0.198 ^b^
Surface area of the left foot (cm^2^)	117.81 ± 21.50	114.43 ± 22.70	0.485 ^b^
Load on the right foot (%)	48.88 ± 4.99	49.31 ± 2.78	0.589 ^a^
Load on the left foot (%)	51.10 ± 4.97	50.69 ± 2.78	0.611 ^a^
Load on the right forefoot (%)	25.75 ± 5.21	23.41 ± 5.56	0.021 ^a^*
Load on the left forefoot (%)	26.57 ± 5.80	25.16 ± 5.89	0.136 ^b^
Load on the right rearfoot (%)	23.09 ± 5.47	26.01 ± 4.22	0.004 ^b^*
Load on the left rearfoot (%)	24.59 ± 4.88	26.60 ± 4.34	0.025 ^a^*
Load on the forefoot (%)	52.33 ± 8.66	46.40 ± 8.16	<0.001 ^b^*
Load on the rearfoot (%)	47.64 ± 8.67	52.42 ± 8.27	0.006 ^b^*

Abbreviations: MS, multiple sclerosis; SD, standard deviation. ᵃ Mann–Whitney U test. ᵇ Student’s *t*-test for independent samples. * Statistical significance at *p* < 0.05.

**Table 3 healthcare-13-01231-t003:** Ankle DF: static analysis in patients with and without MS.

Measurement	Patients with MS n = 42	Patients Without MS n = 42	*p* Value
	Mean ± SD	Mean ± SD	
DF with knee flexed (right)	5.95° ± 4.50	15.45° ± 5.04	<0.001 ^b^*
DF with knee flexed (left)	6.76° ± 4.69	14.90° ± 5.43	<0.001 ^b^*
DF with knee extended (right)	2.69° ± 3.69	8.17° ± 3.41	<0.001 ^a^*
DF with knee extended (left)	3.12° ± 3.83	8.60° ± 3.31	<0.001 ^a^*

Abbreviations: DF, dorsiflexion; MS, multiple sclerosis; SD, standard deviation. ᵃ Mann–Whitney U test. ᵇ Student’s *t*-test for independent samples. * Statistical significance at *p* < 0.05.

**Table 4 healthcare-13-01231-t004:** Logistic regression model predicting MS diagnosis.

	B	Standard Error	Wald	gl	Sig.	Exp (B)	95% C.I. for Exp (B)
Anxiety	2.223	1.016	4.785	1	0.029	9.238	1.260–67.721
Load on the forefoot	0.157	0.068	5.353	1	0.021	1.170	1.024–1.336
DF with knee flexed (right)	−0.535	0.162	10.969	1	<0.001	0.586	0.427–0.804
DF with knee extended (right)	0.509	0.255	3.998	1	0.046	1.664	1.010–2.742
DF with knee extended (left)	−0.407	0.200	4.114	1	0.043	0.666	0.450–0.986
Constant	−2.925	2.928	0.998	1	0.318	0.054	

Dependent variable: Diagnosis of MS (yes/no). Abbreviations: DF, dorsiflexion; MS, multiple sclerosis.

## Data Availability

The data presented in this study are available on request from the corresponding author. The data are not publicly available due to privacy or ethical restrictions.

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
