# Peer review of "Impact of Multiple Sclerosis on Load Distribution, Plantar Pressures, and Ankle Dorsiflexion Range of Motion in Women"

_healthcare, 2025, doi:10.3390/healthcare13111231_

Round 1
Reviewer 1 Report
Comments and Suggestions for Authors
This cross- sectional study compares the impact of multiple sclerosis on load distribution, plantar pressure and ankle dorsiflexion range of motion in Spanish women from Alicante and Murcia. The authors found significant differences in load of right foot and ankle dorsiflexion between MS and normal patients. Although the sample size is relatively small, the authors have conducted a piori samples size calculation to ensure the study is sufficiently powered. My remaining comments as follows:
Abstract: highlight the research gap before the objective statement to give some contact to the readers.
Introduction: in the last paragraph, the authors have highlighted the importance of the study. Please also consider to have a stronger research gap statement to show the readers whether similar study has been conducted or not.
Methods: please add whether informed consent was obtained from the patients.
Sample size: I assume the word precision is actually ‘effect size’. Please insert the citation for effect size at the end of the sentence, not the end of the paragraph.
Statistical analysis: the authors mentioned the use of both Shapiro Wilk and KS test for normality assessment. Please indicate clearly which is the actual test used considering the sample size. Change qualitative variables to categorical variables to avoid confusion. Are there any adjustments to confounders in the analysis?
Results: my major contention is the lack of confounders in the comparisons. For examples, numbers of smokers are different. Could smoking confound the results? What about other comorbidity?
There is also a lack of association analysis between subjects characteristics with dependent variables.
Author Response
Dear Reviewer,
Thank you very much for giving us the possibility of addressing all the questions that arose during the review process. We think that the comments have greatly improved the quality of this observational study. Please find below all the responses in a point-by-point fashion. In the new revised version, the changes are highlighted in red font.
Comments 1:
- This cross- sectional study compares the impact of multiple sclerosis on load distribution, plantar pressure and ankle dorsiflexion range of motion in Spanish women from Alicante and Murcia. The authors found significant differences in load of right foot and ankle dorsiflexion between MS and normal patients. Although the sample size is relatively small, the authors have conducted a piori samples size calculation to ensure the study is sufficiently powered.
Response 1: Thank you for your comment.
Comments 2:
- My remaining comments as follows:
- Abstract: highlight the research gap before the objective statement to give some contact to the readers.
Response 2: We have included a small paragraph in the abstract before the objective statement (lines 23-27).
“Alterations in static plantar pressure distribution serve as important indicators of gait and balance impairments in individuals with Multiple Sclerosis (MS). In addition, the identification of altered patterns of plantar load distribution, along with restricted ankle dorsiflexion, may serve as early markers of postural instability and gait dysfunction in women with MS.”
Comments 3:
- Introduction: in the last paragraph, the authors have highlighted the importance of the study. Please also consider to have a stronger research gap statement to show the readers whether similar study has been conducted or not.
Response 3: The following paragraph has been added (lines 91-100);
“Moreover, sex-specific biomechanical and physiological factors in women may significantly influence plantar pressure, load distribution, and ankle mobility. Characteristics such as a more anterior center of gravity [19], increased knee valgus [20], greater ligamentous laxity [21], and morphological differences in the foot [22] —such as a higher and stiffer plantar arch— directly affect the biomechanics of the foot-ankle complex. In the specific context of MS, sex-related differences in static postural control have also been reported, with women showing poorer performance [23]. Therefore, focusing the analysis exclusively on women allows for better control of these variables and a more in-depth understanding of the functional impairments most representative of this population, which also exhibits a higher prevalence of the disease.”
Comments 4:
- Methods: please add whether informed consent was obtained from the patients.
Sample size: I assume the word precision is actually ‘effect size’.
Response 4: Obtaining informed consent appears on line 132. We've changed “precision” to “effect size” (Line: 124).
Comments 1:
- Please insert the citation for effect size at the end of the sentence, not the end of the paragraph.
Response: We have made the change (line 125).
Comments 5:
- Statistical analysis: the authors mentioned the use of both Shapiro Wilk and KS test for normality assessment. Please indicate clearly which is the actual test used considering the sample size.
- Change qualitative variables to categorical variables to avoid confusion.
- Are there any adjustments to confounders in the analysis?
- Results: my major contention is the lack of confounders in the comparisons. For examples, numbers of smokers are different. Could smoking confound the results? What about other comorbidity?
- There is also a lack of association analysis between subjects characteristics with dependent variables.
Response 5: The following paragraphs has been added (lines 172-189, lines 233-244, lines 246-248);
“Inferential analyses were conducted using the chi-squa.re test for categorical qualitative variables. For quantitative variables, the choice of test depended on the distribution and homogeneity of variances: the Mann-Whitney U test or Kruskal-Wallis test was used for non-parametric data, while the Student’s t-test or ANOVA was applied when parametric assumptions were met.
Relationships between quantitative variables were examined using Pearson's correlation coefficient when both variables followed a normal distribution, and Spearman's rank correlation coefficient when this assumption was not met.
To evaluate the association between the dependent variable (MS) and independent variables while controlling for potential confounders, multiple logistic regression models were constructed. A backward elimination method based on maximum likelihood es-timation was employed, initially including variables that showed statistical significance in the bivariate analysis (p < 0.05).”
“Additionally, associations between participant characteristics and the main outcome measures were explored. BMI showed significant correlations with several outcomes: ankle dorsiflexion with the knee flexed on the left foot (rs = –0.254; p = 0.020), mean plantar pressure in both feet (right: rs = 0.284; p = 0.009; left: rs = 0.331; p = 0.002), plantar contact surface area (right: rs = 0.686; p < 0.001; left: rs = 0.601; p < 0.001), forefoot load on the left foot (rs = 0.229; p = 0.036), and total forefoot and rearfoot load (rs = –0.230; p = 0.035 and rs = 0.225; p = 0.019, respectively). MS clinical phenotype was also significantly associated with mean plantar pressure in the left foot (p = 0.024), with higher values observed in participants with relapsing-remitting MS (144.02 ± 174.95 kPa), followed by those with primary progressive MS (114.50 ± 0 kPa) and secondary progressive MS (103.86 ± 7.76 kPa). No other significant associations were found between age, disease duration, or EDSS and the remaining outcome measures.”
“Finally, multivariate analysis showed a statistical association between MS diagnosis and anxiety, forefoot load, and ankle dorsiflexion in different conditions. The model showed good explanatory power, with a Nagelkerke R² value of 0.75 (Table 4).”
Reviewer 2 Report
Comments and Suggestions for Authors
Review healthcare-3628439-peer-review-v1
The paper titled "Impact of Multiple Sclerosis on Load Distribution, Plantar Pressures, and Ankle Dorsiflexion Range of Motion in Women" presents a clear and methodologically sound investigation into a relatively understudied area of Multiple Sclerosis (MS) - its impact on plantar pressure patterns, load distribution, and ankle dorsiflexion in women. The primary aim of this study was to evaluate differences in static plantar pressure distribution, load bearing, and ankle dorsiflexion range of motion between women diagnosed with MS and healthy controls. The study offers clinically relevant insights that contribute to a more nuanced understanding of postural control and mobility impairments in individuals with MS. Despite its strengths, the manuscript would benefit from a more robust contextualization within the existing literature, a clearer articulation of its novelty, and an acknowledgment of methodological limitations, particularly the absence of dynamic gait analysis.
Introduction: The rationale for focusing exclusively on women should be better developed (lines 52-55), with an extended discussion of known biomechanical and physiological sex differences that may influence sole pressure patterns, load distribution, and ankle mobility. This would help to clarify the limitations on the generalizability of the results to broader populations. Additionally, the manuscript would benefit from a clearer statement of how the study addresses a specific gap in the existing literature (lines 90-92). As it stands, the research appears largely confirmatory; strengthening the claim of novelty by juxtaposing it with previous studies and highlighting what new insights have been gained would enhance the originality and relevance of the article.
Material and Methods: This part is clear, transparent, and reproducible. However, the diagram (lines 137-151) illustrates the configuration of the sole pressure measurement, which would improve methodological clarity and help readers better understand the experimental procedures.
Results: This part is clear, transparent, and reproducible. Tables are comprehensive and statistically annotated. Good use of comparative statistics.
Discussion: The discussion adequately addresses several important factors contributing to gait instability in multiple sclerosis (MS), such as spasticity, muscle imbalance, and emotional factors (lines 204-248). However, the manuscript would have benefited from the inclusion of additional references to existing research on dynamic analysis of soleus pressure in multiple sclerosis, which could have strengthened the rationale for including dynamic assessments in future research (lines 261-263). Furthermore, although the implications for clinical practice are acknowledged (lines 273-282), they remain somewhat general; the authors are encouraged to propose specific rehabilitation strategies or technologies, such as biofeedback systems, targeted strength training, or orthotic interventions - that could be directly based on their findings. Finally, a more critical evaluation of how electromyography (EMG) or kinematic data could complement the current soleus pressure and range of motion results would deepen the biomechanical relevance of the study and suggest avenues for more comprehensive future analyses.
Citations: The majority of citations are current and relevant (2021-2025), covering clinical, biomechanical, and neurological fields. No excessive self-citation was observed, reinforcing objectivity.
Author Response
Dear Reviewer,
Thank you very much for giving us the possibility of addressing all the questions that arose during the review process. We think that the comments have greatly improved the quality of this observational study. Please find below all the responses in a point-by-point fashion. In the new revised version, the changes are highlighted in red font.
Comments 1:
- The paper titled "Impact of Multiple Sclerosis on Load Distribution, Plantar Pressures, and Ankle Dorsiflexion Range of Motion in Women" presents a clear and methodologically sound investigation into a relatively understudied area of Multiple Sclerosis (MS) - its impact on plantar pressure patterns, load distribution, and ankle dorsiflexion in women. The primary aim of this study was to evaluate differences in static plantar pressure distribution, load bearing, and ankle dorsiflexion range of motion between women diagnosed with MS and healthy controls. The study offers clinically relevant insights that contribute to a more nuanced understanding of postural control and mobility impairments in individuals with MS.
Response 1: Thank you for your comment.
Comments 2:
- Despite its strengths, the manuscript would benefit from a more robust contextualization within the existing literature, a clearer articulation of its novelty, and an acknowledgment of methodological limitations, particularly the absence of dynamic gait analysis.
- Introduction: The rationale for focusing exclusively on women should be better developed (lines 52-55), with an extended discussion of known biomechanical and physiological sex differences that may influence sole pressure patterns, load distribution, and ankle mobility. This would help to clarify the limitations on the generalizability of the results to broader populations.
- Additionally, the manuscript would benefit from a clearer statement of how the study addresses a specific gap in the existing literature (lines 90-92).
- As it stands, the research appears largely confirmatory; strengthening the claim of novelty by juxtaposing it with previous studies and highlighting what new insights have been gained would enhance the originality and relevance of the article.
Response 2: The following paragraph has been added (lines 91-100);
“Moreover, sex-specific biomechanical and physiological factors in women may significantly influence plantar pressure, load distribution, and ankle mobility. Characteristics such as a more anterior center of gravity [19], increased knee valgus [20], greater ligamentous laxity [21], and morphological differences in the foot [22] —such as a higher and stiffer plantar arch— directly affect the biomechanics of the foot-ankle complex. In the specific context of MS, sex-related differences in static postural control have also been reported, with women showing poorer performance [23]. Therefore, focusing the analysis exclusively on women allows for better control of these variables and a more in-depth understanding of the functional impairments most representative of this population, which also exhibits a higher prevalence of the disease.”
Comments 3:
- Material and Methods: This part is clear, transparent, and reproducible. However, the diagram (lines 137-151) illustrates the configuration of the sole pressure measurement, which would improve methodological clarity and help readers better understand the experimental procedures.
Response 3: We have added an image to make this part clearer (Figure 1 – Line 167).
Comments 4:
- Results: This part is clear, transparent, and reproducible. Tables are comprehensive and statistically annotated. Good use of comparative statistics.
- Discussion: The discussion adequately addresses several important factors contributing to gait instability in multiple sclerosis (MS), such as spasticity, muscle imbalance, and emotional factors (lines 204-248).
Response 4: Thank you for your comment.
Comments 5:
- However, the manuscript would have benefited from the inclusion of additional references to existing research on dynamic analysis of soleus pressure in multiple sclerosis, which could have strengthened the rationale for including dynamic assessments in future research (lines 261-263).
Response 5: The following paragraph has been added (lines 320-330);
“In this regard, incorporating dynamic assessments of plantar pressure in future research would be highly beneficial. Previous studies have shown that individuals with MS present significant alterations in dynamic plantar pressure parameters during gait. For instance, Keklicek et al. reported disrupted rearfoot loading and delayed load transfer to the forefoot in people with MS, indicating impaired coordination in the gait cycle. Similarly, Jones and van Emmerik found that individuals with reduced plantar vibration sensitivity exhibited increased plantar pressures during walking, particularly in the less sensitive foot. These findings suggest that altered sensory input may contribute to compensatory pressure-loading strategies during gait. Integrating dynamic plantar pressure analysis with static assessments could provide a more comprehensive understanding of postural and gait adaptations in MS and inform more effective rehabilitation strategies.”
Comments 6:
- Furthermore, although the implications for clinical practice are acknowledged (lines 273-282), they remain somewhat general; the authors are encouraged to propose specific rehabilitation strategies or technologies, such as biofeedback systems, targeted strength training, or orthotic interventions - that could be directly based on their findings.
- Finally, a more critical evaluation of how electromyography (EMG) or kinematic data could complement the current soleus pressure and range of motion results would deepen the biomechanical relevance of the study and suggest avenues for more comprehensive future analyses.
Response 6: The following paragraph has been added (lines 349-354);
“Among the evidence-based strategies are eccentric exercises, which have been shown to be effective in improving ankle joint mobility. Likewise, the use of dynamic foot orthoses, textured insoles, and footwear with additional cushioning has demonstrated benefits for balance and gait. In addition, robot-assisted sensorimotor ankle training appears to be a promising tool to enhance strength, range of motion, balance, and locomotion in this population.”
Comments 7:
- Citations: The majority of citations are current and relevant (2021-2025), covering clinical, biomechanical, and neurological fields. No excessive self-citation was observed, reinforcing objectivity.
Response 7: Thank you for your comment.
Reviewer 3 Report
Comments and Suggestions for Authors
Dear Authors,
Congratulations on your study. I believe the research is well conducted and offers valuable insights. However, I would like to offer a few suggestions that may enhance the methodological rigor and the clarity of your findings.
Firstly, one of the primary purposes of calculating an a priori minimum sample size is to ensure that the study includes approximately the appropriate number of participants. However, this estimate is inherently limited, as the actual data and effect sizes ultimately determine adequacy. Therefore, I recommend including a post hoc power analysis to inform readers whether the final sample size was sufficient or if the study may have been underpowered.
Secondly, your analysis focuses on differences in static plantar pressure, load distribution, and ankle dorsiflexion between women with MS and those without. While this is interesting, the rationale for selecting only these parameters is not sufficiently explained. In particular, ankle dorsiflexion is a complex measure that may be influenced by several factors. For example, why was muscle strength of the ankle dorsiflexors not assessed? Simply stating that dorsiflexion was reduced does not clarify the underlying cause—is the restriction due to muscular weakness or a joint limitation? I believe this aspect warrants further elaboration.
I hope these comments are helpful in strengthening your work.
Author Response
Dear Reviewer,
Thank you very much for giving us the possibility of addressing all the questions that arose during the review process. We think that the comments have greatly improved the quality of this observational study. Please find below all the responses in a point-by-point fashion. In the new revised version, the changes are highlighted in red font.
Comments 1:
- Congratulations on your study. I believe the research is well conducted and offers valuable insights. However, I would like to offer a few suggestions that may enhance the methodological rigor and the clarity of your findings.
Response 1: Thank you for your comment.
Comments 2:
- Firstly, one of the primary purposes of calculating an a priori minimum sample size is to ensure that the study includes approximately the appropriate number of participants. However, this estimate is inherently limited, as the actual data and effect sizes ultimately determine adequacy. Therefore, I recommend including a post hoc power analysis to inform readers whether the final sample size was sufficient or if the study may have been underpowered.
Response 2: Thank you for your comments. We share your observations and appreciate the comments. Using the effect sizes observed in the variables that were statistically significant, a post hoc power analysis was conducted. In most cases, the power exceeds 80%, except for the variable "smoking habit," so the sample size used can be considered adequate for detecting the expected effects. However, it would be interesting to conduct additional studies with larger sample sizes.
Comments 3:
- Secondly, your analysis focuses on differences in static plantar pressure, load distribution, and ankle dorsiflexion between women with MS and those without. While this is interesting, the rationale for selecting only these parameters is not sufficiently explained. In particular, ankle dorsiflexion is a complex measure that may be influenced by several factors. For example, why was muscle strength of the ankle dorsiflexors not assessed? Simply stating that dorsiflexion was reduced does not clarify the underlying cause—is the restriction due to muscular weakness or a joint limitation? I believe this aspect warrants further elaboration.
Response 3: The following paragraph has been added (lines 356-362);
“Although this study does not focus on investigating the underlying causes of limitation in ankle dorsiflexion, it has been identified that various conditions can contribute to this alteration, such as soft tissue contractures, joint blocks, and neurological alterations, among others [38]. Once the limitation has been detected in this population, it is necessary for future studies to address the possible underlying causes, with the aim of developing guidelines and protocols that guide personalized therapeutic interventions, considering sex differences.”
Round 2
Reviewer 3 Report
Comments and Suggestions for Authors
Accept